# Assessing the Effectiveness of Green Landscape Buffers to Reduce Fire Severity and Limit Fire Spread in California: Case Study of Golf Courses

Claudia Herbert * and Van Butsic

Department of Environmental Science, Policy, and Management, University of California, Berkeley, CA 94720, USA; vanbutsic@berkeley.edu
* Correspondence: claudiaherbert@berkeley.edu

**Abstract:** Communities looking to improve fire protection may consider incorporating landscape features that 'buffer' the effects of a fire between developed and undeveloped lands. While landscapes such as golf courses, vineyards, or agriculture are already being considered part of this buffer zone, few empirical studies demonstrate causally how well these different landscape features operate as a fire buffer. This research selects golf courses as an example of a possible buffer landscape and proposes methods to test if this buffer alters fire severity and limits fire spread. Using propensity score matching and multiple linear regression, we demonstrate golf courses that burned in California between 1986 and 2020 had a predicted 49% reduction in fire severity relative to otherwise similar vegetated land. This reduction in fire severity is regionally dependent, with the effect of golf course buffering landscapes most pronounced in the North Bay region. For limiting fire spread, golf courses function similarly to hardscaped land uses such as airports, suggesting that irrigation and vegetation management can be effective in creating desired buffering qualities. These methods suggest that artificially created irrigated green zones act as effective buffers, providing de facto fuel breaks around communities, and can be reproduced for other potential buffering landscape features. This study does not advocate for the use of any specific anthropogenic landscape feature, but rather highlights that community-based fire hazard reduction goals could be attained through considering landscape features beyond fuel reduction manipulations.

**Keywords:** WUI; vegetation fire; propensity score matching; fire severity; landscape ecology

## 1. Introduction

Increasingly destructive fire seasons in the Western United States underscore the need for people and communities to proactively plan for living with fire [1–3]. A suite of approaches, ranging from reducing fuels near homes to changing building materials or the spatial configuration of buildings, are considered fire risk mitigation strategies. These mitigation strategies are actions people or communities can take prior to a fire to reduce the harm from a fire event or reduce the chance of ignition [4,5]. Fire risk mitigation strategies can occur across a range of spatial scales, from the individual building to the neighborhood or community level [6]. Regardless of the spatial scales over which these strategies are implemented, their core purpose is to slow or prevent fires from spreading, hopefully reducing fire intensity enough to provide opportunities for fire suppression agencies to safely control fire spread. Such strategies require maintenance to retain the same fire behavior-altering benefits over time [7].

At the community and neighborhood scales, strategic land use planning in the Wildland Urban Interface may be an effective wildfire risk mitigation tactic. Strategic planning includes considering how to arrange different land uses to benefit from changes in fuel availability or conditions. An example is to 'buffer' between vegetated lands and human developments [6]. Proponents of buffering communities point to landscape features such

as orchards, vineyards, parks, and golf courses as landscapes that could interrupt fuel continuity and limit wildfire spread, acting as de facto fuel breaks [8]. A benefit of this approach is that these are often economically generating land uses, so the upkeep necessary to deliver the fire risk benefit is tied to its productive activities; therefore, fire risk benefits happen without policy interventions or public funding, differing from more traditional fuel break maintenance. While the idea of incorporating buffering landscapes seems intuitive, outside of case studies, there is limited empirical evidence across broad spatial scales to quantify the impact of such buffer zones [8,9].

One reason for this lack of evidence is that it is methodologically difficult to quantify buffering effects across a range of geographies. To formalize whether a landscape feature acts as a 'buffer,' we propose that effective buffers should both (*i*) reduce fire severity when burned and, (*ii*) more optimistically, not burn—limiting fire spread, either by not having enough fuel to carry fire or by providing access for suppression. By quantifying these two properties, we can measure a given landscape feature's buffering capacity, allowing it to be evaluated against competing land uses or fire risk mitigation strategies.

Fire severity is a measure of biomass or soil change used to gauge how intensely an ecosystem was impacted by fire [10]. It is directly related to the amount of energy a fire produced, also called fire intensity [11]. When fire intensity is high, direct fire suppression is less effective and carries greater risk for fire suppression crews [12]. Reduced fire severity can indicate areas of reduced fire intensity, possibly providing opportunity for effective fire suppression. However, trying to establish that a landscape reduces fire severity causally requires a control observation to compare outcomes. Using golf courses as an example, identifying a control group can be difficult outside of experimental settings because factors that influence where a golf course is located could also influence fire severity, such as slope, vegetation type, or vegetation moisture. Therefore, establishing that a reduction in severity occurred on a buffer requires identifying statistically similar controls that burned in the same fire or under similar conditions. Fortunately, pre-processing methods such as propensity score matching are an established method for identifying such quasi-experimental controls and have been used in numerous land use policy evaluations [13–16]. With a pre-processed dataset, analysis such as linear regression can be used to predict treatment effect of buffers on fire severity.

Second, an effective buffer should not burn, either because it is not flammable or because it provides access for suppression. If a landscape feature does not carry fire when a fire approaches, it could be considered to limit fire spread. However, limiting fire spread is another challenging concept to test using observational data. One approach could be comparing fire boundaries with buffering landscape features to examine the frequency of certain types of landcovers near a fire edge. However, observing a landscape feature at the edge does not necessarily indicate that the landscape feature had a special property that limited fire spread. Instead, a landscape feature could occur near a fire edge due to chance or due to an unrelated process such as ease of delineating a cartographic boundary near an established edge of a large landscape feature. For possible buffering landscapes that are observed near a fire edge, we could test if there is something special about the type of landscape by comparing it with otherwise similarly shaped and sized vegetated landscapes. This would allow for testing the hypothesis that certain types of landscape management confer advantages in limiting fire spread.

We selected golf courses as a proof of concept for testing the methods developed here to evaluate buffering capacity. In California, golf courses are found throughout the state and have a different vegetation and irrigation intensity than surrounding vegetation. Golf courses are potentially effective as fuel breaks because they are typically at least 150 feet wide in most places and consist of mowed grasses—consistent with fuel break design [17]. Importantly, golf courses are already considered part of 'Buffer Zones' by the National Wildfire Coordinating Group, because they are areas of reduced vegetation separating wildlands to vulnerable residential or business developments [18]. We propose two sets of

quantitative methods for evaluating a given landcover's observed buffering capacity when exposed to a fire, to answer:

1.　Do golf courses alter fire severity relative to similar vegetation?
2.　Do golf courses limit fire spread? How does this compare to other landscape features like parks or airports?

## 2. Materials and Methods

### 2.1. Study Period and Regions

The study region included fires in California that burned between 1986 and 2020 on or adjacent to golf courses. The perimeters of golf courses in California were gathered from OpenStreetMap Overpass Turbo API by querying for "golf course" under the "leisure" category [19,20]. These golf course polygon returns were limited to the state of California, determined by the California TigerLines Shapefile in ESRI's ArcGIS Pro (version 2.7.1) [21,22]. Overlap between golf courses and fires were determined using Cal-Fire California Fire and Resource Assessment Program (FRAP) fire perimeter data [23]. The fire boundaries reduced the 961-golf course dataset from OpenStreetMap in the state to 89 golf courses that had some spatial overlap with fire.

To answer the first question, if golf courses change fire severity outcomes, we examined 22 fires that burned at least a quarter of a golf course (Figure 1). We used the 25% area cutoff to ensure that there would be sufficient golf course area to sample for our statistical procedures. In total, we examined 29 different golf courses. Nine fires burned more than one golf course and three golf courses burned twice in the multi-decade study period, meaning golf courses burned 32 times over the 35-year study period.

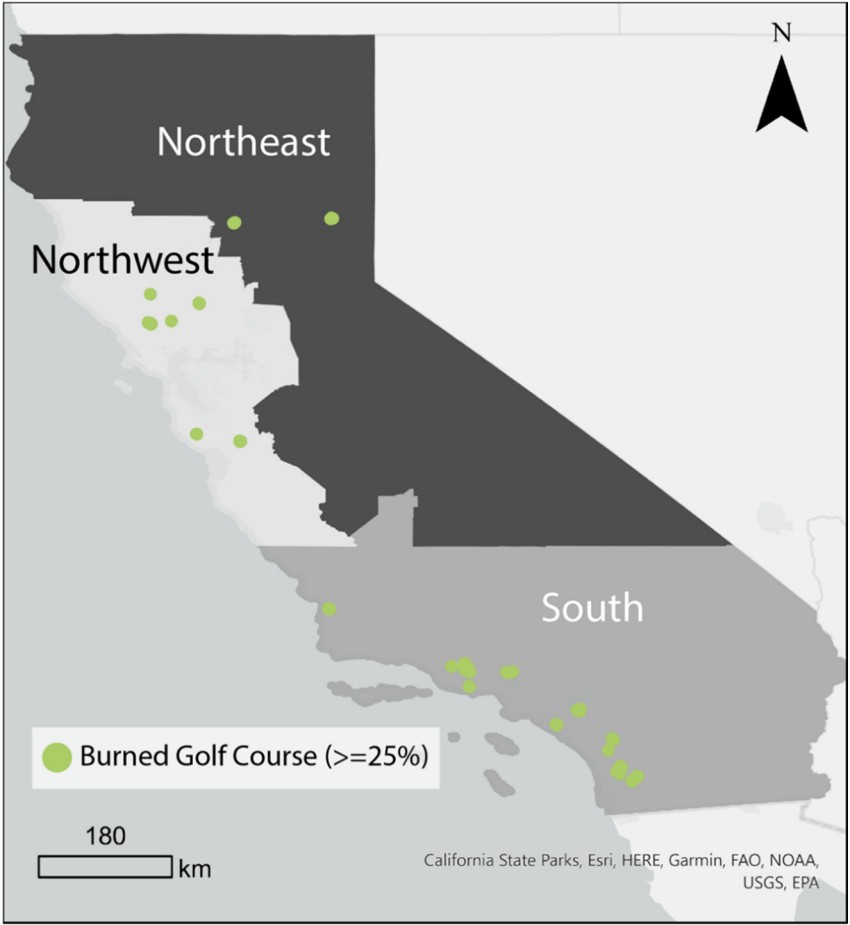

**Figure 1.** Map with the locations of the 29 golf courses in California that had a quarter of their area burned in a fire between 1986 and 2020.

The second analysis, on whether golf courses appear to limit fires spread, spanned 122 cases of golf courses that intersected a fire perimeter between 1986 and 2020. To interpret the effectiveness of golf courses relative to other landscape features, we also looked at parks and airports in California that burned or had some overlap with fire between 1986 and 2020. We limited the park data to be similar in size to golf courses by creating a cutoff of one standard deviation of golf course sizes, or 36 acres to 226 acres. Concerning vegetation management, we assumed that parks of this size would have less water-intensive management than golf courses but may have some fuel management (e.g., grazing, chemical or mechanical removal, prescribed fire). On the other hand, airports are likely managed for more limited vegetation with more hardscape than a golf course or regional park but, such as parks, are unlikely irrigated. This added 121 parks and 49 airports to our study, also queried from OpenStreetMap Overpass Turbo API [24,25].

### 2.2. Fire Severity Data

Fire severity is a measure of how fire intensity affected ecosystems [11]. We approximate this initial impact from fire as changes in vegetation surface reflectance from before and after a fire. Data for fire severity was generated in Google Earth Engine (GEE), which provided access Landsat 5 Thematic Mapper (TM), Landsat 7 Enhanced Thematic Mapper (ETM+), and Landsat 8 Operational Land Imager and Thermal Infrared Sensor (OLI/TIRS) archived imagery [26]. We used GEE to process vegetation fire severity maps for the 22 fires that burned 29 golf courses and generated differenced Normalized Burn Ratio with offset ($dNBR_{offset}$, Equation (1)) maps to calculate severity. The $dNBR_{offset}$ is the average dNBR value from pixels in relatively homogenous, unchanged areas outside the fire perimeter, intended to account for differences in phenology or precipitation between pre- and post-fire images [27]. This GEE script is based on the Parks et al. 2018 paper (corrected in 2021) and used the same cloud masking algorithm, three-sensor harmonizing method, and $dNBR_{offset}$ calculations described in that paper.

$$dNBR_{offset} = ((NBR_{prefire} - NBR_{postfire}) \times 1000) - dNBR_{offset}, \tag{1}$$

The pre-fire date was determined using imagery composited from one and a half months prior to the fire 'alarm date,' recorded in CalFire FRAP. This six-week window ensured that there was enough imagery to create a clear-sky composite of vegetation pre-fire. Similarly, the post-fire images were selected one and a half months post fire 'alarm date.' Some fires in the CalFire FRAP include containment dates that could have been used as the end date of fire, but these dates are not available for all relevant fires and may have more bureaucratic relevance than ecological importance, as these dates reflect when post-fire surveys determined that fires ended [28]. Using the imagery immediately after fire, we calculate an initial burn severity to capture the immediate change from fire, closer approximating fire severity instead of an extended burn severity measure [11]. We expected golf courses that did burn to receive swift management actions, meaning that using imagery too long after a fire would include the management response more strongly than adjacent vegetation. While we are interested in approximating fire severity using remote sensing data, oftentimes, this same technique is used to produce burn severity maps and the terms are used interchangeably [11]. We refer to our produced data as burn severity values for consistency within the remote sensing literature but discuss our results in terms of fire severity.

After maps of continuous burn severity values are produced, burn severity data is often categorized using ground-truth measures of vegetation burn severity. Monitoring Trends in Burn Severity (MTBS) is a concerted multi-agency effort to produce such categorical burn severity maps [29]. Because MTBS was produced for only 15 of the 22 fires relevant to this study, and ground-truth data was not available for all the study sites, we used the raw continuous approximations of burn severity for analysis. As a robustness check, we sampled the categorical MTBS maps for the available 15 fires at the same points as our

continuous burn severity estimates. We used the MTBS categories to develop distributions of continuous burn severity values to interpret our predicted $dNBR_{offset}$ values.

### 2.3. Propensity Score Matching and Linear Regression

Statistically, comparing burned golf course vegetation to vegetation outside of a golf course can lead to biased results because the treated and control observations may be different in ways that are not observed by the researcher. To overcome this, we used propensity score matching (PSM) to analyze nonrandom, observational data by creating a valid set of control and treatment observations [30]. The treatment and control points were first generated using a 100 m grid that considered all points within the burned portion of a golf course perimeter to be 'treatment,' and all non-golf course points within a fire perimeter to be 'control.' Any vegetated pixel within the burned portion of the golf course was eligible for sampling, regardless of whether it was on the fairway or more densely vegetated fairway adjacent areas. The PSM used biophysical and socioeconomic variables we thought would be relevant for identifying non-golf course lands that are otherwise like golf courses, such as the slope, aspect, vegetation moisture, total rainfall, latitude, vegetation type, and average household income (Table 1).

**Table 1.** The variables and their origin used for the PSM and regression analysis. PSM used all the listed variables except for the Burn Severity, which was only used in the regression analysis.

| Type | Name | Description | Spatial and Temporal Resolution | Source |
|---|---|---|---|---|
| Burn Severity | NBR offset | Normalized Burn Ratio Offset using pre-fire dates from 1.5 months pre-fire to 1.5 months post fire | 30 m; 16-day | Landsat 5 TM, Landsat 7 ETM+, Landsat 8 OLI/TIRS [31] |
| Geography | Eastness | Aspect-derived measure of 'east' facing, determined by sine function transformation | 90 m; DEM from 2000 | NASA SRTM [32] |
| | Northness | Aspect-derived measure of 'north' facing, determined by cosine function transformation | 90 m; DEM from 2000 | NASA SRTM |
| | Slope | STRM-derived DEM in GEE to calculate slope | 90 m; DEM from 2000 | NASA SRTM |
| | Latitude | Latitude of each pixel in degrees determined by GEE function | NA | Google Earth Engine |
| Vegetation and Vegetation Moisture | NDMI 6 | Normalized Difference Moisture Index, NDMI = (NIR − SWIR) / (NIR + SWIR), taken from clear-sky composited image between 3 and 6 months pre-fire alarm date | 30 m; 16-day | Landsat 5 TM, Landsat 7 ETM+, Landsat 8 OLI/TIRS |
| | Precipitation | Total precipitation from 1 January–31 March (mm) | 5566 m; daily | CHIRPS [33] |
| | Precipitation 3 | Total precipitation from 1 October–31 December (mm) | 5566 m; daily | CHRIPS |
| | Landcover | Dominant vegetation determined by satellite and ground-truthed data. The landcover closest to the date prior to fire is used. | 30 m; epochs produced for 2001, 2004, 2006, 2008, 2011, 2013, 2016 | NLCD Land Cover [34] |

**Table 1.** *Cont.*

| Type | Name | Description | Spatial and Temporal Resolution | Source |
|---|---|---|---|---|
| Suppression Effort | Median Income | Median household income from the five-year 2018, 2013, 2009 ACS and 2000 Decennial surveys | Variable; 5-year and 10-year | US Census 5-year American Community Survey and Decennial [35] |

We implemented the PSM using matching software for causal inference, MatchIt (version 4.3.0) package in R Studio using the package 'dplyr' to organize data [36–39]. Our model used nearest neighbor matching without replacement and matched points exactly on fire name and landcover to ensure that comparisons about fire severity would be contained by fire and landcover. We used a caliper, or cutoff of maximum difference, of 0.1 to remove any treatment observations that did not have a control observation sufficiently similar. After matching, we have a dataset with a control observation similar to each of the remaining treatment observations, which allows us to use regression analysis to estimate the treatment effect.

To identify the treatment effect on the $dNBR_{offset}$ burn severity outcomes, we then applied multiple linear regression to a dataset consisting only of matched observations. Along with the treatment group, slope, latitude, NDMI from three to six months pre-fire (NDMI 6), total precipitation from October–December (Precipitation 3), median household income, and landcover used in the matching process as explanatory variables, we added a categorical regional variable to indicate whether the fire was in the Bay Area (northwest), North Interior (northeast), or Southern California (south). These regional categories are consistent with Syphard et al. (2021) that found housing pattern and vegetation impact on housing structure loss varied by region of California [40]. This region variable is included in the linear regression as an independent variable and an interaction term with treatment. Adding region to the regression as an interaction with treatment allowed us to calculate the treatment effect on burn severity for each region separately. This allows us to investigate whether the treatment effect of golf courses vary by region.

We implemented the multiple linear regression in the base 'stats' package in R. To interpret the linear regression output, we predicted the average marginal effect using the 'margins' package in R (version 0.3.26) [41]. To predict and plot the regional and treatment interaction term, we used the 'emmeans' package in R (version 1.6.3) [42]. The difference in the predicted treatment $dNBR_{offset}$ from the control group's prediction is the estimated treatment effect of vegetation being managed as a golf course instead of a similar non-golf course. To interpret the continuous $dNBR_{offset}$ predicted values for the regional treatment effects, we use an empirical cumulative distribution function on regionally subset distributions of MTBS categories (unburned, low, moderate, and high) to calculate the predicted values percentile.

*2.4. Measuring How Golf Courses Limit Fire Spread*

For a landscape feature to be an effective buffer, it should both reduce fire behavior and limit fire spread. It is difficult to measure how landscape features limit spread, because a golf course occurring on the edge of a fire perimeter could be the result of a random process or cartographic convenience (using green golf courses to delineate fire boundary) instead of a buffering characteristic of the golf course. To test how golf courses could limit fire spread, we looked at (*i*) how frequently fires stop at golf courses and (*ii*) if this is different than other landscape features, such as parks or airports.

To calculate the frequency of fires stopping at golf courses, we used the ArcGIS Pro (Version 2.7.1) Tabulate Intersection tool to determine the amount of overlap between the OpenStreetMap golf course perimeters and the FRAP fire perimeters [22]. This process included all golf courses that had some overlap or touched the edge of a fire between 1986 and 2020. Golf courses that had close to 0.0% percent overlap with fire represent areas

where the golf course may have limited fire spread. We then used 'ggplot2' package in R to plot the distribution of golf course burned proportions [43]. To help interpret how golf courses may limit fire spread, we recreated the analysis for parks and airports, landscape features that are similar in shape and size to golf courses but have different vegetation management and may exhibit different abilities to limit fire spread. The differences in expected vegetation management between golf courses, parks, and airports is how we test whether an effect of limiting fire spread is present, despite these landcovers being similar in size and shape as landscape features.

To test the differences in the proportion of overlap between the three types of land use, we use a Shapiro–Wilk test to determine that the distributions of proportion burned for the golf courses, parks, or airports are not normally distributed. We then performed a Mann-Whitney-Wilcoxon test to test for similarities in distribution between golf courses and the two alternative landcovers, parks and airports. A Wilcoxon test does not assume a normal distribution of observations and can test for similarities of distributions where the samples are independent [44]. The null hypothesis is that golf courses and parks or airports burn at a similar proportion. If we reject the null, we conclude that the groups are different in terms of the proportion burned when they are near or inside of a fire perimeter. We performed a two-sided Wilcoxon test because we did not want to assume the distributions would always be higher or lower on airports or parks, relative to golf course distributions.

Along with comparing distributions, it may be useful to compare the most frequent burn percentages between golf courses with airports and parks. To do this we implemented a two-sample bootstrap test to sample medians with 2500 replications [45]. This approach allows us to calculate the differences in medians observed between the different landcover samples. If the distribution of the 95% of differences contain zero, then we can say that the medians between the sample are similar.

### 2.5. Robustness Checks

We were concerned that this polygon-overlap approach may depend on the precision of the polygon perimeters; differences in precision between polygons used for golf courses, airports, and parks could lead to erroneous conclusions. As a robustness check, we compared the OpenStreetMap park data with a well-established park dataset, coming from the California Protected Areas Database (CPAD) Units [46]. We assessed agreement using a two-sample bootstrap to compare the median values with 2500 replications and found that there are no statistical differences between medians of the two sources of park data using a Confidence Interval of 95%, which suggested that the OpenStreetMap data sufficiently captured the expected burn proportion from a reputable source such as CPAD.

Moreover, we considered that determining which golf courses, parks, or airports were near fire edge may be sensitive to the precision of a FRAP fire perimeter. We constructed 10 and 20 m buffers around fires to compare how our selection of golf courses, parks, and airports could change if the fire perimeter was slightly different. We found that the number of golf courses, parks, and airports that would have been included in our analysis if the fire perimeter were 10- or 20- meters wider was relatively consistent with our original selection. If the fire perimeters had been 20 m wider, this would have increased the number of golf courses included in the study by a 7 percent difference, added zero additional parks, and added 4.8 percent airports. This exaggerated buffer adding few airports and golf course gave us confidence that the selection of golf courses, parks, and airports were relatively unsensitive to imprecisions in the fire perimeter dataset.

## 3. Results

### 3.1. Quality of Burn Severity and Matched Data

This PSM reduced the mean differences between treated and control observations for the continuous variables included in the linear regression (Figure 2). The caliper and exact matching on landcover and fire name removed 202 treatment observations that did not have a sufficiently similar control observation. This process reduced our dataset

from 363,843 observations with 1572 treatment samples to 2740 observations of matched treatment and control points. The mean differences between the control and treatment datasets improved the most for slope, precipitation between October and December, NDMI three to six months prior to fire, and latitude. Because the PSM resolved differences between the average control and average treatment for all continuous variables, we felt that this matched dataset was well-suited for linear regression and prediction.

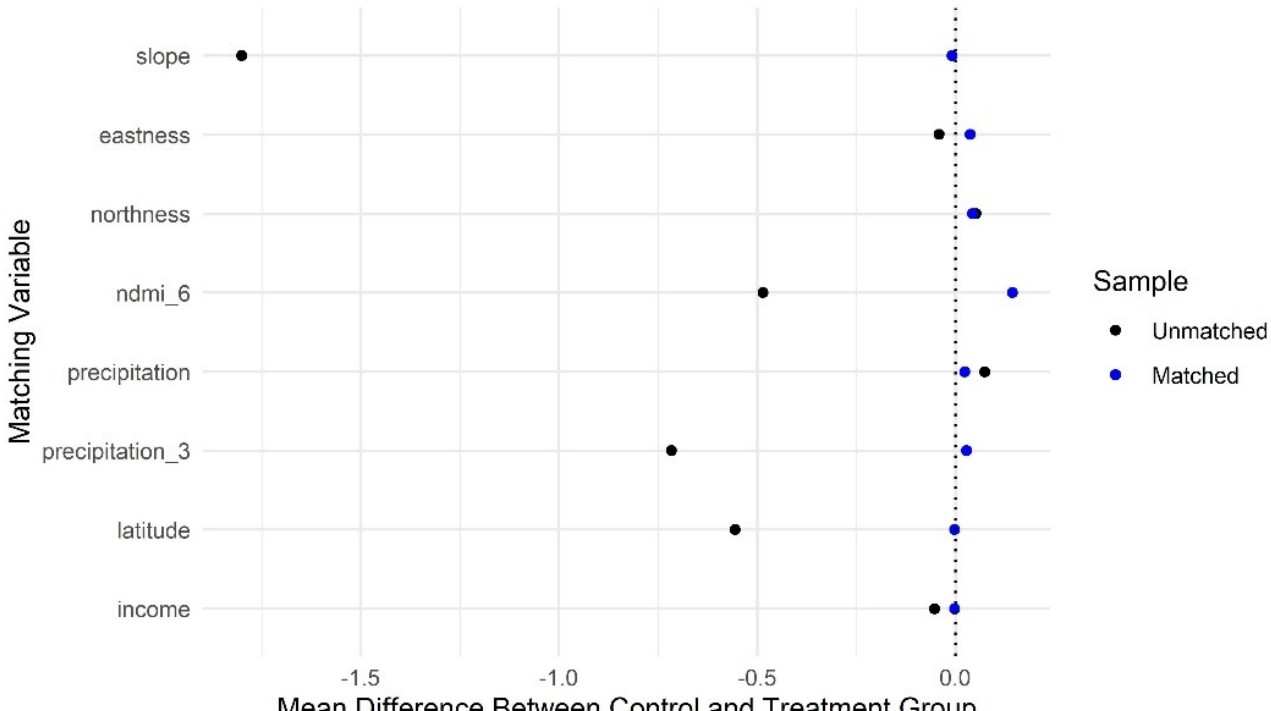

**Figure 2.** Love plot for the standardized mean differences for the continuous matching variable prior to and after the PSM generated by using the 'cobalt' package in R [47]. Exact matches on categorical data such as landcover and fire name have been removed from this plot.

### 3.2. Linear Regression and Predicted Treatment Effect

According to our regression and prediction analysis, the treatment effect of a golf course across all California regions reduced dNBR fire severity by an average of 65 units, compared to otherwise similar vegetation (Figure 3). A lower $dNBR_{offset}$ for golf courses indicates less change in surface reflectance post fire. This difference between the predicted control (163.11) and treatment (97.96) is a 49.91% difference. Because there is no overlap between the predicted 95% confidence intervals, we conclude that this treatment effect difference between golf courses and otherwise similar non-golf course vegetation is statistically significant. However, because $dNBR_{offset}$ values are most meaningful within the context of a single fire, and these predictions include fires across California over decades, interpreting fire severity outcomes is less straightforward.

Based on the regional and treatment interaction term predictions, we found that the treatment effect does vary by region in California (Figure 4). We found the largest treatment effect in the northwest region. Based on the northwest predicted treatment average of −41.49 to the predicted control at 99.90, we used the MTBS regional distributions to assess where in the burn severity categories these predictions fall (Table 2). We suspect that the treatment effect in the northwest is roughly a whole burn severity class. The northwest treatment prediction falls at the 10th percentile of the 'unburned' category, and the northwest control prediction is above the 63rd percentile for 'unburned' and around the 23rd percentile for 'low.' Because there is no overlap between the predicted 95% CIs, this treatment effect is statistically significant.

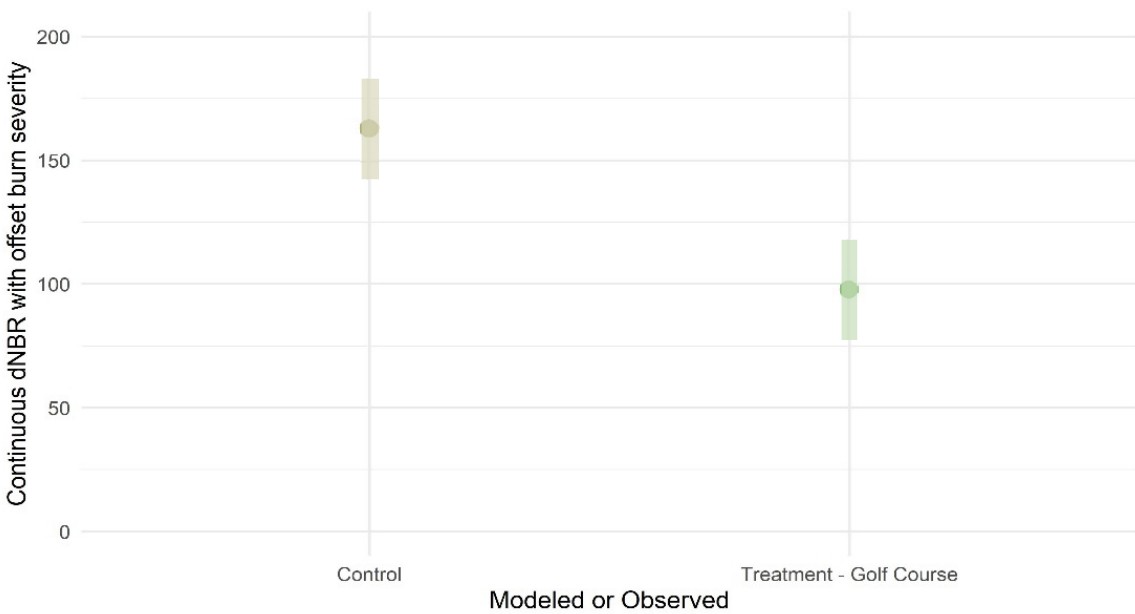

**Figure 3.** Predicted fire severity values for non-golf course vegetation (control, left) and golf courses.

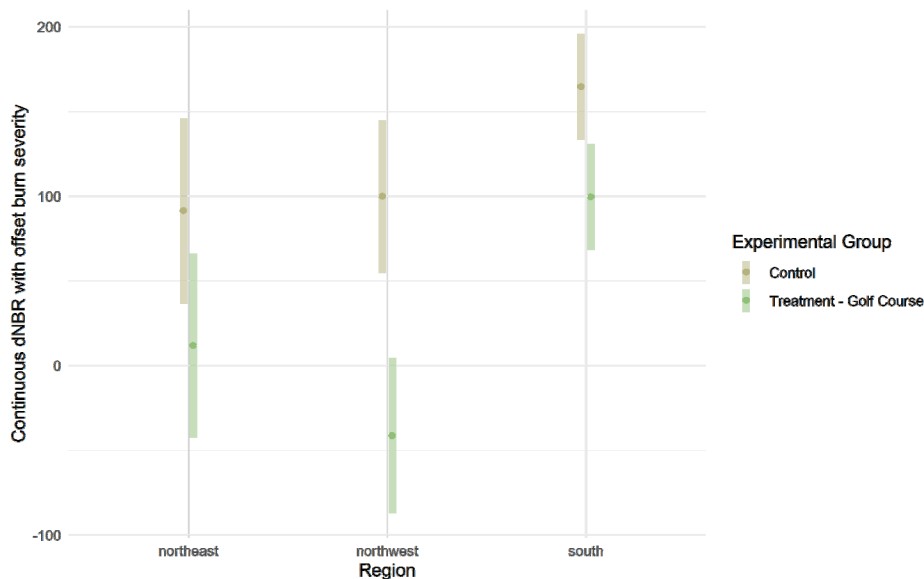

**Figure 4.** Predicted fire severity values for golf courses, golf course edges, and non-golf course vegetation by region.

**Table 2.** Predicted values and where the predicted dNBR$_{offset}$ values fall on corresponding MTBS distributions.

| Region | Group | n. | Predicted | SE | LCL | UCL | Unburned | Low | Moderate | High |
|--------|-------|-----|-----------|-------|--------|--------|----------|--------|----------|--------|
| northeast | Cont. | 142 | 91.36 | 27.88 | 36.69 | 146.02 | 52.17% | 31.51% | 12.03% | 8.00% |
| northeast | Treat. | 142 | 11.70 | 27.68 | −42.58 | 65.97 | 18.45% | 10.64% | 4.16% | 3.05% |
| northwest | Cont. | 278 | 99.90 | 23.00 | 54.81 | 144.99 | 63.91% | 23.18% | 1.722% | 0.12% |
| northwest | Treat. | 278 | −41.49 | 23.00 | −87.17 | 4.19 | 10.52% | 0.57% | 0% | 0% |
| south | Cont. | 950 | 164.59 | 15.93 | 133.36 | 195.81 | 54.39% | 31.94% | 12.42% | 12.17% |
| south | Treat. | 950 | 99.44 | 15.93 | 68.22 | 130.66 | 44.44% | 19.75% | 6.14% | 7.81% |

The south region, similarly, has no overlapping 95% CI, suggesting these differences are statistically significant. Predicted dNBR$_{offset}$ values for the south treatment group,

99.44, and control group, 164.59, were the highest of the three regions. When we compare these predicted values to the distribution of MTBS categories, we see that these predicted values are still falling towards the middle of the unburned or low-severity fire distributions, despite the higher raw predicted dNBR$_{offset}$ values.

The northeast predicted and controls do have some overlap in their CIs, but using a two-sided student's T-test, we find that these distributions are still significantly different at a 0.05 *p*-value cutoff. The northeast control group is burning at the 52nd percentile for 'unburned' and the 31st percentile for 'low' suggesting that it is somewhere between those groups. The northeast treatment is burning at the 18th percentile for the unburned severity and 10th percentile for low severity, suggesting it is mostly unburned. This distribution comparison between treatment and controls is very similar to the south distributions for MTBS. We believe that both regions have a treatment effect where golf courses are burning at lower severities, but it is not clear that this effect necessarily translates to a burn severity class difference.

### 3.3. Golf Courses Limiting Fire Spread

When a fire burns near a golf course, golf courses most frequently burn less than 10% of their area (Figure 5). The observed distribution of how much of a golf course, park, or airport burn exhibits bimodal peaks; most burning completely or hardly at all. Because the total number of golf courses, parks, and airports differ, to facilitate comparisons between the features, we produced a smoothed density estimate (Figure 5, panel b). This smoothed density estimate makes the bimodal distribution more apparent and reveals a right skew for the airport and golf courses and a left skew for the park dataset. This means that golf courses and airports tend to burn closer to zero when near a fire, whereas parks found near a fire are more frequently entirely contained within the fire perimeter.

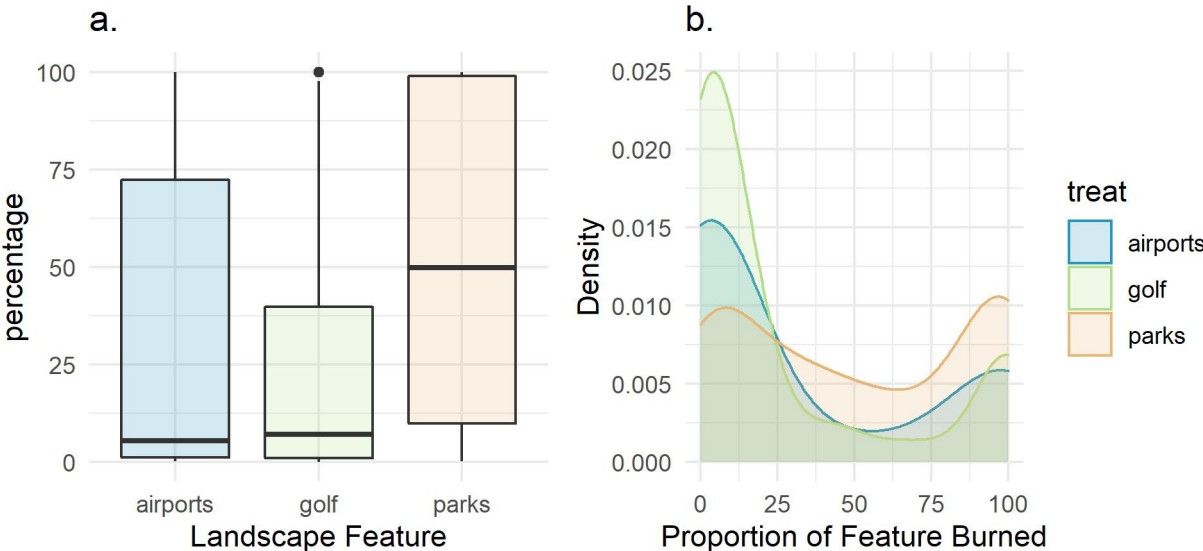

**Figure 5.** Boxplot (**a**) and smoothed density estimate (**b**) distribution of proportion of land burned of the golf, airport, or park polygons.

Using a 5% significance level, the Shapiro–Wilk test indicated that the proportion of burned areas for golf courses, parks, and airports were not normally distributed. This is consistent with the bimodal distribution observed in the histograms. The Wilcoxon test (Table 3) found golf courses and parks were significantly different, so we accept the alternative hypothesis, that the golf courses and parks are different. Conversely, the golf courses and airports test were not statistically significate at this threshold, indicating evidence for the null hypothesis, that the distributions are equal.

**Table 3.** The Wilcoxon test statistic and *p*-value interpretation for the landcover percentage burned comparisons.

| Comparison | Test Statistic | Significant (0.05 Cutoff) |
|---|---|---|
| Golf Course—Airport | 3551 | No |
| Golf Course—Park | 4996 | Yes |

Using the two-sample bootstrap, we found that the differences in the median percentage of golf course burned was not distinguishable from the airport median (Figure 6). Parks, on the other hand, did not have a 95% confidence interval that contained zero, indicating that the median differences between the percentage of golf and parks burned are not similar. These bootstrapped median differences are consistent with the results of the Wilcoxon test, where golf courses are like airports in how they burn but parks are not.

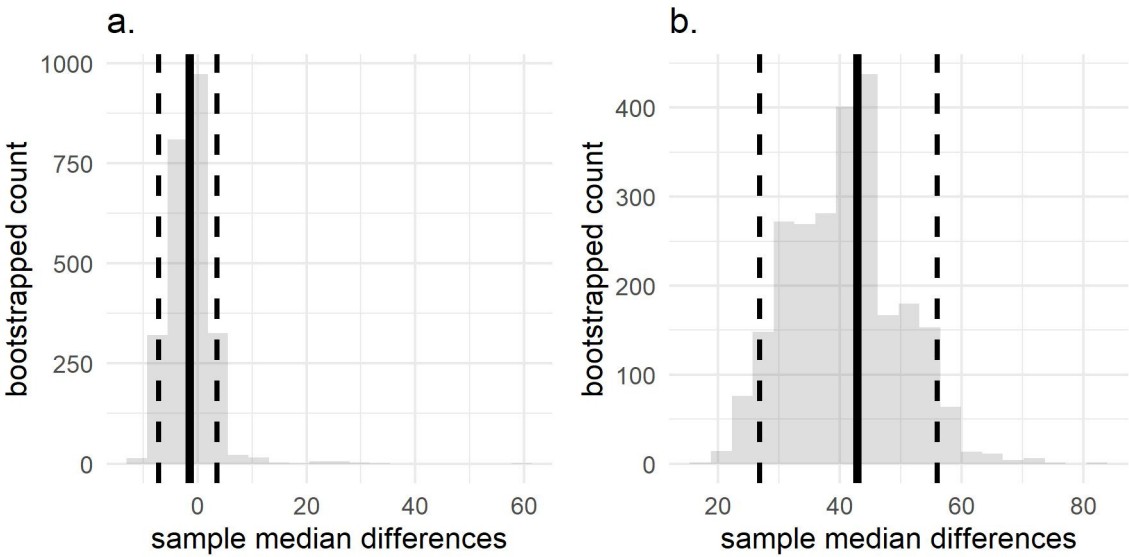

**Figure 6.** Results of the two-sample bootstrap to compare the median differences between (**a**) golf courses and airports and (**b**) golf courses and parks based on median percentage of land feature burned.

## 4. Discussion

Choosing to promote buffer zones around communities is a resource decision. Land use planners and communities should be able to access empirical data to assess a land use's buffering capacity to weigh tradeoffs in making decisions about how to allocate limited resources such as water, fire suppression efforts, or undeveloped land. Here, we provide empirical evidence for the effectiveness of golf courses as fire buffers across California along with the methods for others to recreate this analysis for other landcovers.

Our findings suggest that when a golf course burns, it tends to burn at a reduced severity relative to similar vegetation. Our regression analysis found that this amounted to a 49% difference between golf courses and non-golf course controls. This demonstrated reduction in fire severity fulfills the first criteria we define as 'buffering capacity.' If the reduced fire severity is the result of reduced fire intensity, then these landscape features may provide buffers where suppression crews could safely access and more effectively protect adjacent communities. Because dNBR$_{offset}$ is a metric best used to interpret fire or burn severity within an individual fire, these predicted values are meant to illustrate the direction and significance of golf courses treatment effects. Importantly, there are regional differences in the size of the fire severity treatment effect. We found evidence that the northwest region had the largest treatment effect, possibly reducing burn severities an entire MTBS category, from low severity to unburned. The south and northeast regions

still had statistically significant treatment effects, but it is less clear that these differences translate to a whole burn severity category reduction.

One limitation of this research is our reliance on using fire severity as an approximations of fire behavior. Ultimately, for fire suppression and community protection, fire behavior metrics such as fire intensity are more meaningful. Advances in producing fire intensity measurements from thermal heat signatures, captured by satellites such as MODIS, VIIRS, or GOES, show promise for estimating Byram's Fire Intensity [48,49]. Our proposed approach could be adapted to incorporate fire intensity estimates instead of fire severity. However, a challenge inherent in a thermal-based fire intensity approach is that passive thermal energy is recorded at a coarser spatial resolution than spectral data (e.g., 250 m$^2$ versus 30 m$^2$ pixel sizes), so it may not be sufficient for studying smaller landscape features such as golf courses. Moreover, because golf courses still burn infrequently, averaging less than one golf course over this 35-year study period, a longer archive of imagery was useful for creating a dataset with sufficient observations to compare across regions and a range of vegetation. Future research to improve this method could incorporate field measurements of fire intensity and connect the field measurements with the high spatial-coverage of satellite-derived fire severity. This field data is not available for our historic dataset and requires the ability to collect data during a fire. Alternatively, instead of looking at the fire severity impacts, researchers could also examine other adverse fire consequences such as structure damage from fires.

Despite these shortcomings, the robust treatment effects we observe across all of California and within the region-specific treatment effects suggest that our approximation of fire severity is sufficiently capturing the golf course treatment effect. While our use of initial fire severity is intended to limit the role of post-fire recovery from influencing our results, the effect of swift recovery on more intensely managed landscapes such as golf courses may be influencing our findings.

Through comparing the proportion of golf course area that burns or is adjacent to fires, we found that the low proportion of burning suggests evidence that they act as buffers, limiting fire spread. Golf courses exhibited this low proportion of area burned similar to other landcovers, such as airports, that we expected to be effective at limiting fire spread given the hard-cover and regular vegetation maintenance airports receive. Golf courses did not burn like similarly sized parks, consistent with our expectations of the role of management, and suggesting that our way of approximating this true absence of golf courses burning is valid.

Despite evidence that golf courses may be effective buffers, both reducing fire severity and limiting fire spread, there still may be limitations in their effectiveness that are not apparent in our analysis. Our work relies on observations from historic fires, yet many of the most recent destructive fires in California have been notably wind-driven and may not be well represented in our observational dataset [1]. During these wind-driven fires, golf courses may not provide a wide enough buffer to protect structures from wildland fires, failing to provide effective buffer capacity [50]. This is where considering what other types of 'buffering' landscapes could be combined to create wider buffer matrix could be useful.

Being able to quantify the relative effectiveness of a land use allows planners, communities, and emergency responders to identify key parts of their landscapes that should be maintained or augmented as part of fire preparedness. The effect we report from the linear regression and polygon analysis likely captures some effect of additional fire suppression resources used on these landscapes, rather than solely a physical advantage. For example, golf courses may have more extensive fire suppression activities around them than a similarly vegetated landscape that is not generating similar economic activity per acre. We believe that this factor is why it is important to not overstate the role that vegetation management pre-fire alone contributes to our findings. Still, even if we are measuring the effects of emergency management responses to protect certain landscapes, it is an important step forward in the planning and fire risk literature to quantify the role of placing different land uses around human developments as part of a community-wide fuel treatment and

fire preparedness plan. Beyond California or golf courses, other regions and land uses may have anecdotal evidence supporting a fire buffering effect. The methods outlined in this paper could be used to study whether this buffering effect is observed in other regions for golf courses or other land uses.

Along with suppression during a fire, pre-fire management irrigation is likely different between an open-space park and a golf course. The reported benefit golf courses may have over similar non-golf course vegetation during a fire comes at the cost of the additional irrigation, herbicide, energy, and labor used to manage these land uses. These are fixed inputs where the benefit of reduced fire behavior is only realized during a fire. Along with the direct costs associated with these inputs and reduction in wildlife habitat, there are other social and resource costs when directing limited resources such as water to irrigating a golf course in an arid ecosystem. Finally, any of the observed benefits connected to the physical properties of the fuels on the golf courses are a byproduct of the management. If the management changes, the benefits observed may also change. One benefit of economically generating landscapes such as agriculture or golf courses for land use planning is that they are revenue streams to support their continued management outside of fire related public grants.

Communities or planners considering incorporating managed green landscapes as fire buffers might envision it as part of a resilient community design for living with fire. However, concepts like resilience are difficult to operationalize. Choosing to send resources towards maintaining a landscape like a golf course may provide specific resilience to the threat of fire while not being compatible with other sustainability goals. Resilience is not just the inverse of vulnerability and across different scales of resilience, some interventions might have tradeoffs [51]. For example, maintaining golf courses as fire buffers may increase community resilience to fire and, at another timescale, if the resources needed for maintaining the golf course worsen the local effects of climate change or drought, it could contribute to increased wildfire threats. Tradeoffs between resources and resilience across scales are core to land use planning and management [51]. While we were able to estimate the benefit green landscape buffers might provide, some of the tradeoffs or costs associated with their maintenance might be harder to quantify or lack common measures. Thus, a community resilience assessment for incorporating managed landscapes as fire buffers should consider the incommensurable tradeoffs associated with this land use and resource decision [52]. Understanding where these green landscape buffers should be maintained will largely be context and region specific, and we hope that our outlined method and software will be one tool for decision makers to use to make choices suitable for their circumstances.

## 5. Conclusions

Golf courses can act as a fire buffer around communities, potentially reducing community level risk of wildfire. While anecdotal evidence of golf courses acting as buffers already exists in different regions, using empirical methods we were able to study golf courses in or near fires in California to robustly estimate this buffering effect. We found two main results which indicate golf courses may be effective buffers. First, relative to similar vegetation outside of golf courses, golf course vegetation burns at lower severity. Second, based on the higher frequency with which golf courses appear near fire edges instead of entirely within a fire, we demonstrate that golf courses appear to limit fire spread. Taken together, this evidence suggests that golf courses can be part of a community land use plan to limit wildfire risk. Golf courses in California are just one example of a type of landscape feature which could be incorporated as part of a community buffer zone. We hope that the methods and the software demonstrated in this research will be adapted to study fire buffering effects for other land uses or regions. Ultimately, we hope these tools will help communities plan for fire and weigh the resource tradeoffs associated with the maintenance of those potential buffers.

**Author Contributions:** Conceptualization, C.H. and V.B.; methodology, C.H. and V.B.; software, C.H.; formal analysis, C.H.; data curation, C.H.; writing—original draft preparation, C.H.; writing—review and editing, V.B. and C.H.; visualization, C.H.; funding acquisition, V.B. All authors have read and agreed to the published version of the manuscript.

**Funding:** We thank Lotusland Investment Holdings, Inc./Bohn Valley, Inc. for a financial gift which supported this research.

**Institutional Review Board Statement:** Not applicable.

**Informed Consent Statement:** Not applicable.

**Data Availability Statement:** Publicly available datasets were analyzed in this study. This data can be found here: https://github.com/claudiaherbert/FireBufferCapacity (accessed on 18 February 2022).

**Conflicts of Interest:** The authors declare no conflict of interest. The funders had no role in the design of the study; in the collection, analyses, or interpretation of data; in the writing of the manuscript, or in the decision to publish the results.

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
