# Peer review of "Assessing the Effectiveness of Green Landscape Buffers to Reduce Fire Severity and Limit Fire Spread in California: Case Study of Golf Courses"

_fire, doi:10.3390/fire5020044_

Round 1

Reviewer 1 Report

Line 94 has a typo. add the word "as". For the international reader, the term park should be clarified, especially as there is reference to protected areas in the text. A park can be a highly managed play area, with swings, it could be a sports field or it could be a highly vegetated area. In the light of differences between Golf Courses and Parks (but not airports), this could be a point for clarification. Please note that in Aust, golf courses are already used as buffers, albeit from anecdotal evidence. This research lends support to this. Also note that in many countries, golf courses may be interpreted as meaning areas of heavy vegetation, as opposed to fairways. 

Reviewer 2 Report

Review on Ms. Ref. No.: fire-1624773

Golf courses reduce fire severity and limit fire spread in California

Claudia Herbert and Van Butsic

Recommendation: Accept after minor revision

  1. Summary

The current research paper is focused on the reduction of fire severity and limitation of fire spread in California. The main attention here is paid to the golf courses as examples of a possible buffer landscape and the paper proposes methods to test if this buffer alters fire severity and limits fire spread. However, the golf courses are already considered as part of ‘Buffer Zones’ by the US Wildfire Coordinating Group, the aim of the authors is to evaluate if their buffering capacity is big enough or not. The methodology contains impressive software decisions with the usage of different Remote Sensing and GIS map-creating technics. The results are well-supported by the data presented and the reviewer agrees with your findings. A novel and good research paper!

  1. Major issues

No suggestions for major issues.

  1. Minor issues
  • An interesting note about the authors' writing style needs to be clarified. In almost every sentence there is the personal pronoun "we". On the one hand, many authors are reluctant to write the first person singular, believing that this is a kind of immodest and that it shifts the emphasis from the subject of the article to the author itself. On the other hand, the reviewer also thinks that there are a number of cases in which the use of the pronoun "we" is preferable to the impersonal form. In this regard, the reviewer will be interested to know the opinion of the authors on this topic.
  • Figure 1 is recommended to be in bigger size (and better quality) and proper color consideration because in the current condition the label and the green dots are not legible enough.
  • However the Conclusion is not a 100% mandatory requirement for this journal, the reviewer strongly recommends adding a Conclusion section with the major findings of the research briefly summarized, unloading the contents of the Discussion section.
  • Check one more time the reference list following the journal instructions. Some typos appeared in lines 490, 492. Use this template doi:prefix/suffix.

  1. Opinion

The manuscript represents very good research with an interesting approach, novel technics, and huge efforts. The reviewer strongly believes that it will be accepted after minor revision. Congratulations for the good work to the authors and good luck to the Ph. D. student.

Reviewer 3 Report

In this study, the authors investigated the effect of golf courses on fire severity reduction and fire spread limitation using remotely sensed data and other corresponding spatial data to perform analyses comparing the differences between golf courses, airports, and parks. This investigation may contribute to the understanding of potential role of golf courses related to land planning for disaster-reduction and preparation. Overall, this manuscript is well written and the findings of this study is meaningful with scientific rigorousness. However, there are some fundamental flaws that I need to indicate to the authors for their consideration.

  • First, since the study mainly determined the reduction effects by comparisons of three land use types- golf courses, airports, and parks that may likely have very different vegetation composition inside (the proportion of tree, shrub, and lawn, or even hard pavement) vegetation may play important roles in fire event in addition to irrigation and management, I suggest that the authors should seriously consider the differences of vegetation composition among the three and may adjust the research question 1 (are they really similar in vegetation? Probably not. Or, evidences are needed to support that they are similar).
  • Second, related to the earlier comment, I suggest that the authors to also talk about other two landscape types in Introduction in addition to golf courses for the readers to understand the lines of comparison. Currently, there are too little information for the environmental characters of airport and parks in the background context.
  • At last, in the discussion, regarding the main findings, there is basically a synopsis for the analytic results but is in lack of the elaboration for, such as, possible reasons that cause the results or other in-depth argument closely link to the local situations in California.            

Reviewer 4 Report

This paper attempts to argue for anthropogenic land features e.g. golf courses to act as fire breaks. They base their argument off a TNC report and another report; both of which I’m not sure if they are peer-reviewed (its just not clear). That aside, one can make the argument for golf courses to be fire breaks. But you can do the same for a playground a parking lot, or an agricultural monoculture. A monoculture has no ecological benefit, neither does a playground, or a golf course, but of them serve a purpose in society that makes our world function. I do not think we can advocate for an anthropogenic feature, created to benefit a small community to act as a useful fire break. I don’t argue against having golf courses for people to enjoy their sport, but I don’t think that can be considered as a viable fire break, even if they are. That is not something I will advocate for. We can say the same thing about a parking lot, but we can’t have parking lots in the middle of a tropical dry forest to act as a fire break. Golf courses are a poor selection of landscape feature for this study. Researchers could consider something that actually serves a purpose in the landscape - something society needs to function e.g. roads that also act as fire breaks but actually connect people. I also disagree completely that golf courses are like airports. Airports connect the world. Golf is played by a select community for recreational purposes. I will not comment on anything else on this paper because the foundation of this study is flawed.

Round 2

Reviewer 4 Report

I don't believe this work is appropriate to be part of the fire ecology community. There is no scenario where an anthropogenically derived feature acts a fire break in this century. Perhaps there is a different pitch on this story, perhaps more correlative than causal, but to say that an anthropogenically derived feature - that has such a tremendous amount of conservation flaws - plays a role in controlling fire - in one of the lyric systems in the world, will be a major breakthrough for anthropogenically derived features to expand even further. Consider what that means for the future landscape. Think a bit about the pitch of your study, because a golf course is not a fire break that belongs in the landscape, even if it acts like one. 

Author Response

I appreciate the concerns of Reviewer 4 about the implications of this work. We feel that this is an important academic contribution because communities and land use planners should have ways of estimating potential fire buffer effects, instead of relying on anecdotal evidence. We do not suggest that the resources necessary to maintain the altered vegetation on golf courses, which is likely necessary to observe a buffering effect, is negligible.